# An Evaluation of the Popularity of Australian Native Bee Taxa and State of Knowledge of Native Bee Taxonomy Among the Bee-Interested Public

**DOI:** 10.3390/insects16111149

**Published:** 2025-11-10

**Authors:** Kit Prendergast

**Affiliations:** 1School of Molecular & Life Sciences, Curtin University, Kent Street, Bentley, WA 6102, Australia; kitprendergast21@gmail.com; 2Centre for Sustainable Agricultural Systems, University of Southern Queensland, Toowoomba, Darling Downs, QLD 4350, Australia

**Keywords:** *Amegilla*, Australian native bees, bees, blue banded bees, conservation, science communication, scientific literacy, social media, taxonomy, *Tetragonula*, wild bees

## Abstract

This study aimed to determine the Australian public’s taxonomic knowledge of Australian native bees and identify the relative popularity of species to identify potential flagship or gateway species, as well as taxa that are under-represented. Asking “What is your favourite Australian bee species?” to members joining the Facebook group, it was found that of those answering, only one-quarter provided a species or common name of a species, rather than a group, and less than 10% provided a scientific name, indicating a large gap in taxonomic literacy. Euryglossinae, a diverse, specialised, endemic subfamily, were not mentioned. The most popular were *Amegilla* (“blue banded” and “teddy bear” bees), which may offer a gateway into native bee biodiversity. Our results indicate that greater taxonomic education is required, as the public has poor scientific literacy, which can hamper conservation efforts. Concerted attention is required to raise awareness of under-represented taxa among the public.

## 1. Introduction

The public is becoming increasingly aware of a pollinator crisis (a so-called “bee apocalypse”) and is also a key player in addressing the threats that are causing declines in bees [1,2]. However`, there is often a gap between public knowledge of a topic`, and the science behind it [3,4]. One identified yet little-known threat to insects is a “taxonomic shortfall” and a lack of awareness about the insects that are threatened [3]. This is especially prevalent for bees, where attention is focused on the European honeybee *Apis mellifera*, which is a non-threatened, widespread species, yet attains much publicity due to its economic importance [5]. Not only is the well-known *A. mellifera* an introduced species, and not at risk of extinction, but they also differ greatly in their niche from indigenous bees and thus serve as a poor umbrella species [5], and can even harm native bees [6,7,8], and a focus on helping honey bees can hinder native bee conservation [9,10,11]. The attention of honey bees, whilst drawing attention to pollination services, has done little to advance the advocacy and conservation of native bees [12].

In Australia, there are over 1700 described species [13], with hundreds more yet to be scientifically described [14,15,16]. Australia also has endemic families and subfamilies of bees, which represent an important part of the country’s evolutionary history and endemic biodiversity, whose conservation is therefore important.

Identifying where the public lacks knowledge about indigenous bee taxa, to guide future engagement and educational strategies, can be an important tool for bee conservation, especially because citizens can play a key role in creating habitat for bees (e.g., [17]) and securing protection for species through legislation and habitat protection (e.g., [18]).

Gateway species are those that can aid in awareness of the biodiversity of a broader taxonomic group, being the “gateway” into community awareness [19,20]. Flagship species are high-profile, charismatic, or ambassadorial species that act as symbols, capturing the attention of stakeholders and the wider public, to galvanise interest or action for biodiversity, conservation issues, campaigns, and the wider conservation movement [21,22]. Despite insects representing the bulk of terrestrial fauna diversity, and a publicised insect apocalypse, an insect species is rarely deployed as a flagship species [11,23]. Yet this represents an important strategy for biodiversity conservation because of the public interest flagship species generate, engendering public attention and advocacy for the myriad, less-conspicuous, and undescribed invertebrates [22]. Some examples include *Danaus plexippus* (Monarch butterfly) [24], *Lycaeides melissa samuelis* (Karner blue butterfly) [23], and *Bombus affinis* (Rusty-patch Bumblebee) [25]. Australian examples include *Agrotis infusa* (the Bogong Moth), *Dryococelus australis* (Lord Howe Stick Insect), *Attacus wardi* (the large saturniid Atlas Moth)*, Ornithoptera richmondia* (Richmond Birdwing) [22], and *Synemon plana* (Golden Sun Moth) [26].

Given the diversity of species and taxonomic shortcomings, selecting one or a few native bee species as gateway species can be used to encourage interest in native bees [19]. Selecting flagship species can be used to raise awareness about native bees, champion their cause, and in conservation campaigns. Such a strategy—selecting gateway and/or flagship bee species—could be an effective strategy in light of the under-appreciated crisis facing native pollinators and pollination services in Australia [27].

The aims of this research were as follows:Evaluate bee taxonomic literacy among Australian citizens who were interested in bees;Identify which Australian bee species emerged as the favourite(s) among the public, which can therefore be used as a gateway and/or flagship species;Identify which bee taxa were under-represented and therefore require more outreach and science education.

## 2. Methods and Materials

From 17 October 2021 to 31 July 2022, data on Australian citizens’ favourite bee was collected via an anonymous questionnaire shared via Facebook as part of the required questions that must be answered to join a private Facebook group. To not bias the results, respondents were not aware that the question was for the purpose of determining Australian citizens’ favourite bee nor the extent of their knowledge on native bee taxonomy. Participants were people requesting to join the Facebook group “The Buzz on Wild Bees”, created by Dr Kit Prendergast in August 2016, which at the time of the questionnaire, had approximately 10–11 K members. Existing members were not asked the question, but rather people who were requesting to join the group, which ensured that the findings were not affected by length of duration individuals had been part of the group, and to attain a picture of the taxonomic knowledge and popularity of Australian native bees amongst the general bee-interested population who had not been exposed to native bee educational and taxonomic information whilst being in “The Buzz on Wild Bees” Facebook group. The question was open-ended and simply asked, “What is your favourite Australian bee species?” No identifying information was recorded, simply the answer to the question by Dr Kit Prendergast and the two other moderators of the group, Tamara Venables and Chris Hoy Poy. All answers were then tallied up for each species/group of species. Note that when tallying responses for a given taxa, answers with two answers provided were given a point each.

## 3. Results

Between 17 October 2021 and 31 July 2022, 579 citizens who joined “The Buzz on Wild Bees” Facebook answered the joining question (Appendix A). Of these people, there were a number of responses where no species was provided, including one N/A, one “thought this was a honeybee page” [sic], one “TBC”, and two “did not know there was more than one species” [sic]. There was also a large number who expressed that they were “new and learning” and therefore did not provide a favourite bee species (50 people), “unsure” (66 people), and 7 people “didn’t have one”. Seventeen people said “all”. There were also ten people who did not state a name of a bee and we were unable to discern what taxon they were referring to: “a calm one” (one), “alive ones” (one), “northern bees” (one), “the one living in my plant” (one), “the ones living in my retaining wall” (one), “the ones that pollinate my fruit” (one), “the little small native one” (one), “black” (one), “native bees” (one), “the blue one” (one), “burrowing” (one), “Australian native” (one), “the black bees that have no stripes” (one). One person also answered “hornet”. One person answered “Amarillo”, which is not a name, scientific or common, of any bee as far as we are aware.

Although the question asked for a favourite Australian native bee, there were also ten people who answered with non-Australian bees. A total of 419 Australian native bee taxa were provided as answers from 404 participants (some participants provided more than one taxon in their answer) (Appendix A). The vast majority of respondents used common names, most of which did not refer to a species. Of the 419 answers that referred to Australian native bees, just 57 (13.6%) provided a scientific name, and of these, only 32 (7.6%) were scientific species (genus species) names (Appendix A).

Whilst the question asked for a favourite bee species, only 54 (12.9%) answers were either a (valid) scientific or common name for a species, rather than a group of species (Appendix A).

Grouping together common names and scientific names, the “*Amegilla*” group emerged as Australia’s favourite native bee taxon, with 294 votes (Figure 1). Within the *Amegilla* group, the “blue banded” (most *Notomegilla* and *Zonamegilla* species) (272 votes) were more popular than the *Asarapoda* species (19 votes). The next-most popular taxon was Meliponini, with 68 votes (Figure 1). Where genera or species could be distinguished within Meliponini, *Tetragonula* (40 votes) were more popular than *Austroplebeia*, represented only by *A. australis* (5 votes). The *Megachile* group and the *Xylocopa* group were third, with 19 votes and 21 votes, respectively (Figure 1). Interestingly, despite comprising a very small fraction of the Australian bee fauna, cleptoparasitic bees (largely represented by *Thyreus*, but also one vote for a *Coelioxys* as “cone-waisted cuckoo”) received ten votes (Figure 1). Other taxa received only a few votes: Hylaeinae (3), Halictidae (2), and Allodapini (2) (Figure 1). Apart from Hylaeinae, no other colletid taxa received any votes, nor did Stenotritidae, a family endemic to Australia.

Looking at only valid species names answers, *Tetragonula carbonaria* was the most popular (19 votes), followed by *T. hockingsi* (12 votes), *Austroplebeia australis* (5 votes), *Xylocopa aerata* (“Metallic Green Carpenter Bee”), and *Amegilla cingulata* and *Amegilla bombiformis* (3 votes each).

## 4. Discussion

This research addressed the aims of evaluating the bee taxonomic literacy of the Australian public, determining the popularity of particular species that could hold promise to serve for outreach about native bees as flagship or gateway species, and identifying underrepresented native bee taxa that warrant concerted outreach strategies. Our findings in summary were as follows:(a)At least in terms of the stated favourite bee species, there were gaps in taxonomic knowledge, in terms of scientific nomenclature, taxonomic rank in relation to the question of favourite *species*, and over-representation of certain taxa.(b)“Blue banded” *Amegilla* and *Tetragonula carbonaria*, were the most popular taxa among bee-interested Australian citizens. *Amegilla*, in particular those in the subgenera *Zonamegilla* and *Notomegilla*, emerged as popular native Australian bees among the public, and are good candidates for flagships.(c)Colletids and Stenotritids were extremely under-represented and therefore require more outreach and science education. This is especially concerning since, unlike the other families and genera, these taxa are endemic to Australia.

We discuss our findings in more detail below, revealing the implications of our findings.

### 4.1. Poor Taxonomic Knowledge

Our results suggest the general population of bee-interested citizens have poor knowledge of bee taxonomy. For example, even with this modest sample size of bee-interested people, 10% of people felt they did not know enough about native bees to even provide an answer to their favourite Australian bee species. Many used common names, which obscure taxonomic identity and effective communication about biodiversity [28]. Many respondents also did not answer what their favourite Australian bee species was by providing the name of a species, but rather a group of species which may not map onto a taxonomic category, e.g., “blue banded bee”. This is a crucial lacuna that must be addressed. Without being able to effectively communicate, we cannot communicate about the conservation, resource needs, or threatened status of any given species [28].

It is also evident that much of the public does not know about, or does not find attractive, the vast majority of Australian native bee species. No one indicated that a species in the subfamily Euryglossinae or any of the Neopasiphaeineae was their favourite, yet of the currently described species, these groups collectively constitute over 40% of Australia’s bee biodiversity (Euryglossinae: 404 species; Neopasiphaeineae: 307 species) [13]. Only one halictid was stated as a favourite species, despite halictids being very common bees in anthropogenic habitats. The reasons why these taxa were not included can only be speculated at present, but warrant further investigation. For halictids, it is likely due not to a lack of encountering them, but rather that most *Lasioglossum* are relatively “dull”, being black and non-descript [29]. Regarding Neopasiphaeineae, whilst genera like *Leioproctus* include species that are commonly encountered, many are similar in this respect to *Lasioglossum* in terms of lack of bright colours and striking patterns. *Trichocolletes*—an Australian endemic genus—are beautiful and relatively large, but are specialists and rare in anthropogenic habitats, especially those dominated by exotic plants. In contrast to these taxa, many Euryglossinae have bright colours and patterns. The lack of popularity of the Euryglossinae may stem from their small size (anywhere from a mere 2 mm long, and all under 10 mm in length). Unlike Meliponini, which are also small, they are solitary and cannot be kept, nor are they effective pollinators [29]. Being highly specialised (oligoleges) [29], many of these small bees forage in the canopy of Myrtaeace trees and therefore are less visible than Meliponini, and they are less encountered in anthropogenic habitats dominated by exotic plants such as traditional urban gardens [30,31] or agricultural fields [32]. This further underscores that greater awareness of these specialised, endemic bees is needed.

The relative popularity of certain taxa cannot be explained on the basis of biogeography. Meliponini, despite being small and black rather than large and bright, are absent from the lower half of Australia (mid-west to southwest WA, across through South Australia, into the lower half of New South Wales, ACT, Victoria and Tasmania). In contrast, Euryglossinae, Halictidae, and Neopasiphaeinae can be found across the whole of Australia [33]. The relative popularity also cannot be explained by phenology: in many regions, Amegilla are only active in spring and summer, which is typical of most native bees. At the species-level, whilst *Amegilla cingulata* and *Tetragonula carbonaria*, which are not distributed across Australia [33], received votes, species that occur across Australia and are active for relatively long periods, such as *Lasioglossum urbanum*, *Lipotriches flavoviridis*, *Leioproctus plumosus*, *Hylaeus nubilosus*, *Megachile aurifrons*, and *Megachile tosticauda* received no votes [33]. Euryglossinae species can also be one of the most abundant species in an area [34,35,36], and therefore, relative rarity cannot explain the lack of representation. It is therefore likely that the relative popularity stems from both media representations and a good public narrative, along with the ability to ‘keep’ Meliponini, and the larger and brighter traits of Amegilla.

Interestingly, the cleptoparasitic bees, being relatively uncommon by virtue of being parasitic, and comprising a small subset of the Australian bee fauna, received a number of votes. This hints that to raise awareness and popularity among native bees, a compelling story is needed [37,38] that focuses on building a “character” for native bee taxa. Whilst “cuckoo bees” received much attention when featured in media such as that by *Australian Geographic* in “The neon cuckoo bee is a shiny parasite” [39], there has been scant media attention on Neopasiphinae or Euryglossinae, which may explain their lack of representation in the favourite Australian bee species votes.

It is evident that scientific names are memorable and recognisable based on people using the correct names for *Tetragonula carbonaria* and *T. hockingsi*, which, like the vast majority of Australian native bees, do not have common names. These species received more votes than votes for “common name” species, and thus it is clear that the Australian public can and do learn and use scientific names if they are introduced to them and they become standard. Educators should desist from using common names, promote the use of the scientific names to increase scientific literacy, make taxonomy accessible and interesting to the public at large, and promote a more universal and democratic means of communicating about biodiversity [28,40,41].

This is the first Australian study to explore the lack of public knowledge of this country’s endemic bee biodiversity, and demonstrates that the Australian public suffers from ‘Species Awareness Disparity’ (SAD) in bees, which is defined as the failure to appreciate the significance of wild bee species and the inability to distinguish between individual species [4]. Professional entomologists are aware of this issue in other countries. This includes Southeast Asia [42], which revealed a startling lack of awareness of bee biodiversity, as well as how greater awareness of bees was associated with greater positive opinions on the intrinsic value of bees, further supporting our conclusions on the importance of exposing and educating the public to bee biodiversity to ensure public support in their conservation. In both India and the USA, many people lack awareness and the ability to identify of non-honey bee species [12,43], and likewise in Germany, students in grades 5–7 demonstrated a poor awareness of wild bees [4], whilst in Spain and Brazil, there is a notable lack of knowledge regarding bee diversity among primary and university students [44,45].

### 4.2. Why Do the Public Love “Blue Banded Bees” so Much?

There has been research, mainly focusing on vertebrates, about what makes a species a good “flagship” species among the public. One aspect is being “charismatic”—brightly coloured, easy to observe, and fairly common across a range of habitat types, and regularly encountered, creating familiarity [23]. The various species of “blue banded” *Amegilla* conform to these characteristics: they have colourful blue bands (Figure 1), are relatively large compared with many native bees and are thus easy to observe, are common across a range of habitat types, occur across Australia, and as they are generalists and will feed on exotic plants, are regularly encountered [46].

It should be noted that of the *Amegilla,* only two subgenera (*Notomegilla* and *Zonamegilla*) include species with blue bands, and not all species in these genera have bands that are blue, e.g., Amegilla *aeruginosa* [46]. There are also non-*Amegilla* bees with blue bands, e.g., some *Nomia* [47].

### 4.3. Popularity of Meliponini, Especially Tetragonula carbonaria and T. hockingsi

The Meliponini emerged as the next-most popular bees, and for the species-level results, *Tetragonula carbonaria* and *T. hockingsi* were the favourites. Like *A. mellifera,* Meliponini are eusocial, produce honey, and can be kept in hives [48]. The popularity of *T. carbonaria* and *T. hockingsi* likely stems from how much research has been put into these species, which means they can be kept, used for pollination and beekeeping, and have an economic value [49,50,51]. There has been a lot of outreach to the public in the form of popular books (e.g., [52,53,54,55]), presentations and workshops (e.g., https://australiannativebee.org.au/events, accessed on 7 September 2025), and media about these species as well.

### 4.4. Popularity of Megachilids and Xylocopa

Whilst the respondents had poor taxonomic knowledge of these groups, megachilids (“leaf-cutters” or “resin bees”) and *Xylocopa* (“carpenter bees”) were also popular. For the megachilids, this may stem from how people can “keep” them on their properties across Australia through bee hotels, which have become a popular feature and, for better or worse, are available to buy at most garden stores [56,57,58,59]. This allows these bees to be seen where people live, and as such, the familiarity and personal connection with these bees may contribute to their popularity [60]. Viral videos on Facebook, Twitter, and Instagram of megachilids emerging from cocoons may also play a role [61], e.g., this [61] video posted on Facebook by The Center of Biological Diversity of *Megachile* bees emerging from their leaf-cocoons received 18 K likes in just ten days: https://www.facebook.com/CenterforBioDiv/videos/478486344290570, accessed on 17 September 2022. Social media networks such as Entomemology may be used to create “memes” to promote less-attractive species [62].

For the *Xylocopa*: these are relatively large, charismatic bees, which again increases the likelihood of people encountering them. *Xylocopa aerata* was one of the most popular species. This is a large bee, and is extremely visually attractive, with a metallic green body. Its congener, *Xylocopa bombylans*, did not receive in votes, despite it being very similar visually. The popularity of *X. aerata* likely stems in part from its plight being raised across multiple media platforms in the wake of its precarious status on Kangaroo Island (e.g., [63,64,65,66,67,68,69]).

### 4.5. Amegilla as a Gateway, but Not a Flagship

Given the clear popularity of *Amegilla* among the public, species in this genus could serve as appropriate flagships for native bee conservation [23]. Indeed, they would fit the definition provided by Barua et al. of an insect flagship species as “an invertebrate species or group that resonates with a target audience and stimulates awareness, funding, research and policy support for the conservation of invertebrate diversity.” [70] They are solitary, like many native bees [71]. The genus *Amegilla* can also be used to demonstrate diversity, with three different subgenera and 36 described species [13]. This diversity within the genus should, however, be emphasised, and the different distribution and ranges of the species communicated so that the public does not think all *Amegilla* have blue bands and are equally common. As non-honey producing, non-hive species, they can also be used to promote people to care for indigenous biodiversity based on its intrinsic, rather than economic use value [72]. Their effectiveness at buzz pollination can also be used to encourage an understanding of the specialised plant–pollinator interactions in Australia, of which the introduced *Apis mellifera* cannot replace in this respect [73]. Where *Amegilla* may fall short in serving as an “umbrella” species [21] is that they are generally resilient to urban landscapes, and are polylectic, foraging on a range of host plant species, including exotic flora, and they are also ground-nesters [46]. Consequently, conserving habitat for *Amegilla* will not automatically secure populations of oligoleges, or cavity-nesting bees.

Further actions are required to raise awareness about different *Amegilla* species (i.e., not one “blue banded bee”) to encourage greater appreciation of diversity, taxonomic literacy, and opportunities for interaction and connection with species local to a region [28,74].

*Amegilla cingulata* was recently heralded as the 2024 Australian Insect of the Year, and thus the results presented here align with showing that this species is popular among the Australian public from a much larger voting pool [75]. This present study, however, differs from the Australian Insect of the Year survey, as our question was open-ended, which enabled determining both people’s preferences and taxonomic knowledge, rather than limiting the options to a set of six pre-selected insect species.

*Amegilla* species may be better suited to serve as “gateway” species—the public clearly has readily fallen in love with *Amegilla*, and through their popularity, this can act as a starting point to stimulate conservation awareness and actions for native bees. Through learning about these bees, we hope that this can be used in communication and conservation strategies to open a “gateway” into learning about native bees and their ecologies as a whole.

### 4.6. Selecting a Flagship

Due to the generalist and adaptable nature of many *Amegilla*, it may be more advisable to select flagships that not only can garner public attention, but also a species for which conservation actions are able to protect a wider diversity of other threatened, less conspicuous native bee species. Different flagships may be best selected in different regions of Australia, but the popularity of *Xylocopa* and *Megachile*, their vulnerability to habitat loss (reliance on suitable trees), and often strong association with native flowers mean that such species can be selected by scientists and science communicators to garner interest and push for conservation action that benefits other native bee species. What is needed to aid this is good photographs, exposure, and a good story—it is likely these three aspects help explain the popularity of *Amegilla cingulata*, *Xylocopa aerata*, and *Tetragonula* among the public. Facebook groups such as “The Buzz on Wild Bees” can aid in such a strategy.

### 4.7. Future Research

The sample size was relatively small, but this was restricted by the number of people requesting to join ‘The Buzz on Wild Bees’ Facebook group over the duration that the collection of answers took place. This strategy was conducted to reduce the bias in answers from those who had joined the group and had learned from the content the admin (a native bee expert and scientific educator) was posting. It would be interesting to conduct the survey again with members who had been an active part of the group (i.e., reading posts) to assess if their exposure to native bee taxonomy and a greater biodiversity of native bees would alter their literacy on native bee taxonomy and their choice of what Australian native bee is their favourite.

This was therefore a pilot study on a sample of people interested, but not necessarily knowledgeable about, native bees. It should be noted that findings on bee taxonomic knowledge may have been even more limited if the sample had been a random selection of the Australian public at large (as opposed to people who were interested in joining a wild bee Facebook group), as it is likely that bee knowledge (and indeed, scientific literacy) is poorer in the general population. It is also acknowledged that future studies on the Australian public’s favourite bee species and their taxonomic knowledge would be better implemented within a research framework and a greater number of questions. The number of questions was limited to ensure that people joining ‘The Buzz on Wild Bees’ were not put off from joining due to needing to answer a large number of questions. Intentionally, it was not presented to those joining that the survey was targeted at finding out people’s favourite bee species or their taxonomic knowledge, as this could bias results, e.g., people rallying behind or being influenced by “voting” for a particular taxon (e.g., [76]), or looking up bee scientific nomenclature prior to answering. Future research could also ask more questions to better unpack why people like a particular bee species/taxon, e.g., aesthetics, seeing it in their garden, conservation status, relationship to services such as honey provision, and other demographic questions such as level of educational attainment. Further research into people’s knowledge of bee taxa and their attraction to or against certain taxa may also shed light on why two major taxonomic groups, Euryglossinae and Neopasiphaeinae, received no votes.

It is concerning that an entire group of endemic, largely oligolectic, bees—the Euryglossinae—received no votes, nor did any of the three species listed as Threatened (Critically Endangered) on the EPBC Act: *Hesperocolletes douglasi*, *Leioproctus douglasiellus*, and *Neopasiphae simplicior* (Colletidae and Neopasiphaeinae). Greater public awareness and education campaigns about these under-represented, but most at-risk species, are urgently required. Strategies to address this shortcoming include selecting these species as future candidates for #InsectoftheYear, social media campaigns, representation in articles and websites about native bees, with emphasis on the local and indigenous nature of these species to promote personal connection, and, importantly, inclusion in school curricula, emphasising narrative techniques and showcasing the cute, colourful and harmless nature of these species [77,78]. Research suggests that whilst much of the population is ignorant when it comes to insect diversity, simply being exposed to fascinating “unknown” and unfamiliar insects can incite interest [79], and media portrayals and opportunities for interactions are important [74], as is education on identification [80]. I also recommend educating children by including native bees in school curricula and educational and entertainment materials, and providing hands-on learning activities and encounters with native bees [4,81].

Improving citizens’ understanding of taxonomy is not a mere academic exercise; it is important for increasing scientific literacy [82], fostering environmental responsibility [45], and improving communication about biodiversity [28]. Because successful native bee conservation efforts require public support, including an understanding of the state-of-knowledge among non-experts [12,83] is also important. The findings presented here mirror those in the United States, where the bee-interested public is nevertheless uninformed and has a poor understanding of the diversity of native bees [12]. As with these authors, I agree that the success of the public’s involvement in “saving the bees” will improve with a greater understanding of the bees that require saving, given that different native bee taxa have different requirements, and misguided efforts may do little to support native bees in need [12].

## 5. Conclusions

This research has revealed both the relative popularity among the Australian public of Australian native bee taxa and the paucity of taxonomic knowledge. Future research is required to determine if educational strategies can increase taxonomic literacy. In addition, a number of hypotheses are presented about why certain bee taxa were relatively popular or unpopular, which require sociological research to support or disprove these hypotheses. Such results can then aid in raising the profile of lesser-known bee taxa, which may be in critical need of greater public, political, and financial support.

## Figures and Tables

**Figure 1 insects-16-01149-f001:**
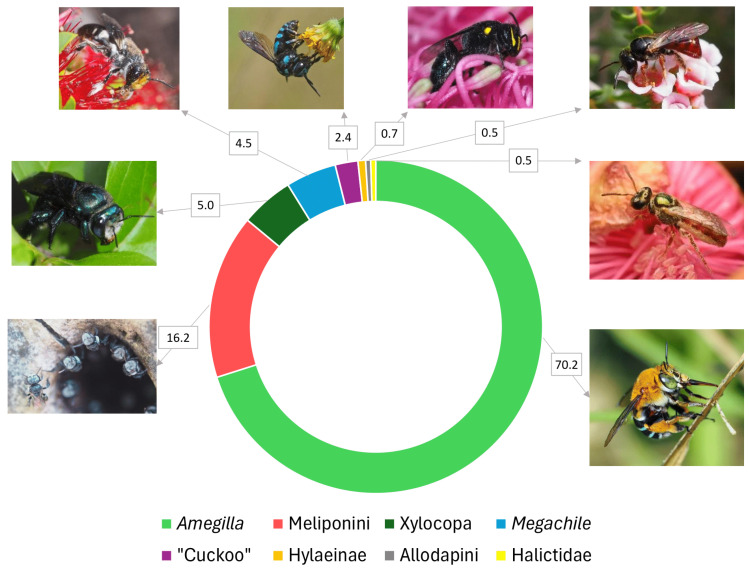
Popularity of Australian native bee taxa based on responses from the public joining ‘The Buzz on Wild Bees’ Facebook group, who were asked, “What is your favourite Australian bee species?”. Photos from left to right: *Amegilla* (“blue banded bees”) represented by *Amegilla cingulata*; Allodapini (which may colloquially referred to as “reed bees”) represented by *Exoneura sp.*; Halictidae (which may colloquially referred to as “sweat bees” or “furrow bees”) represented by *Lasioglossum (Homalictus) dotatum* (syn. *Homalictus dotatus*); Hylaeinae (which may colloquially referred to as “masked bees”) represented by *Hylaeus (Macrohylaeus) alcyoneus*; cleptoparasitic bees (sometimes called “cuckoo bees”) represented here by *Thyreus nitidulus*; *Megachile* (which may colloquially referred to as “leaf-cutter” or “resin bees”) represented by *Megachile (Shizogmegachile) monstrosa*; *Xylocopa* (“carpenter bees”) represented by *Xylocopa (Lestis) aerata*; Meliponini (which may colloquially referred to as “sugarbag bees” or “stingless bees”) represented by *Tetragonula carbonaria*. All photographs by Dr Kit Prendergast, except the photo of *Lasioglossum (Homalictus) dotatum,* photographed by Krystle Hickman. Numbers refer to percentage of responses, rounded to one decimal place.

## Data Availability

All data is available as online Appendix A.

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
