# Peer review of "An Evaluation of the Popularity of Australian Native Bee Taxa and State of Knowledge of Native Bee Taxonomy Among the Bee-Interested Public"

_insects, 2025, doi:10.3390/insects16111149_

Round 1

Reviewer 1 Report

Comments and Suggestions for Authors

Personally I am in favor of approaches that include the general public in studies on (insect) species knowlegde and species awareness. Although most general outcomes and conclusions of this study have been previously shown by quite a few research groups, this study can be considered as a relevant and meaningufl case study for Australia, a continent with a wild bee fauna of its own.

It is unfortunate that only few people of the Facebook group took part in the survey and the author obviously took no measures to improve respondent numbers. Hence, only the data obtained was subsequently analyzed. 

There are two aspects which would improve the paper and can be dealt with on basis of the current data set. 

(1) By using e.g. the Australian Faunal Directory or the Atlas of Living Australia which provide distribution maps for taxa the author could correlate the vote numbers with regional abundances and investigate whether species that are restricted to only some regions are less mentioned. Maybe people recall spectacularily looking species more than "normal" ones? A second path for augmenting the data analyses could be times of the year where species are easily spotted. Moreover, readers who are not familiar with the Australian fauna could follow the species discussions more easily,

(2) The author states that species knowlegde does not reflect taxonomy knowledge. It the latter required for the first? If so, the author could elaborate on this fact make suggestions as to how this can be achieved. 

Although I am not a native speaker, the following sentence (lines 323-325) needs editing:

We acknowledge that are results on bee taxonomy may have been even more limited if this was a random selection of the Australian public at large, where it is likely that bee (and indeed, scientific literacy) is far poorer in the general population.

Author Response

Personally I am in favor of approaches that include the general public in studies on (insect) species knowlegde and species awareness. Although most general outcomes and conclusions of this study have been previously shown by quite a few research groups, this study can be considered as a relevant and meaningufl case study for Australia, a continent with a wild bee fauna of its own.

Response: thank you

It is unfortunate that only few people of the Facebook group took part in the survey and the author obviously took no measures to improve respondent numbers. Hence, only the data obtained was subsequently analyzed. 

Response: I need to emphasize that it was not people who were part of the Facebook group, but rather people who were requesting to join the group during the period of 17 Oct 2021 to 31 July 2022. This was to ensure that the findings were not affected by length of duration individuals had been part of the group, but rather a picture of the taxonomic knowledge of the general bee-interested population. I intend to conduct a follow-up survey of whether people who had joined the group, and consequently been exposed to a greater array of information about native bees, had improved taxonomic knowledge. The question was posed as one of the questions required for people to answer to join this Private Facebook group, and therefore the respondents were not aware that it was to discern taxonomic knowledge or identify a favourite Australian native bees, which may have biased the results.

I have now made this more clear in the Methods.

There are two aspects which would improve the paper and can be dealt with on basis of the current data set. 

  • By using e.g. the Australian Faunal Directory or the Atlas of Living Australia which provide distribution maps for taxa the author could correlate the vote numbers with regional abundances and investigate whether species that are restricted to only some regions are less mentioned. Maybe people recall spectacularily looking species more than "normal" ones? A second path for augmenting the data analyses could be times of the year where species are easily spotted. Moreover, readers who are not familiar with the Australian fauna could follow the species discussions more easily,

Response: I have now included the following, citing the ALA: The relative popularity of certain taxa cannot be explained on the basis of biogeography. Meliponini, despite being small and black rather than large and bright, are absent from the lower half of Australia (mid-west to southwest WA, across through South Australia, into the lower half of New South Wales, ACT, Victoria and Tasmania). In contrast, Euryglossinae, Halictidae and Neopasiphaeinae can be found across the whole of Australia. The relative popularity can also not be explained by phenology: in many regions Amegilla are only active in spring and summer, which is typical of most native bees. At the species-level, whilst Amegilla cingulata and Tetragonula carbonaria, which are not distributed across Australia, received many votes, species that occur across Australia and are active for relatively long periods such as Lasioglossum urbanum, Lasioglossum dotatum, Lipotriches flavoviridis, Leioproctus plumosus, Megachile aurifrons and Megachile tosticauda received few to no votes. Euryglossinae species can also be one of the most abundant species in an area, and therefore relative rarity cannot explain the lack of representation.

It is likely therefore that the relative popularity stems from both media representations, along with the ability to ‘keep’ Meliponini, and the larger and brighter traits of Amegilla.

(2) The author states that species knowlegde does not reflect taxonomy knowledge. It the latter required for the first? If so, the author could elaborate on this fact make suggestions as to how this can be achieved. 

Response: I am afraid I don’t understand the question here.

Although I am not a native speaker, the following sentence (lines 323-325) needs editing:

We acknowledge that are results on bee taxonomy may have been even more limited if this was a random selection of the Australian public at large, where it is likely that bee (and indeed, scientific literacy) is far poorer in the general population.

Response: It should be noted that findings on bee taxonomic knowledge may have been even more limited if this was a random selection of the Australian public at large (as opposed to people that were interested in joining a wild bee Facebook group), as it is likely that bee (and indeed, scientific literacy) is poorer in the general population.

Reviewer 2 Report

Comments and Suggestions for Authors

The study summarizes results from an online survey of people interested in wild bees. The author utilized a Facebook group called “The Buzz on Wild Bees” to ask joining members what their favorite Australian bee species is. The format allowed for open-ended answers. The survey results found limited taxonomic and biodiversity knowledge among survey takers. Of 400 people who provided answers only 19 provided a scientific name, and the majority provided a group rather than a species, underlying the lack of native bee knowledge among survey takers. The most popular group by far were the “blue banded” bees, indicating a potential flagship group for conservation. However, the three critically endangered bees in were not mentioned by any survey participants, highlighting a gap between public knowledge and conservation priorities. Further, the study identifies species or groups of species which were popular among participants, and explores possibilities for designating flagship species.

            The study is effective in communicating that native bee taxonomic knowledge is lacking among the Australian public, even among a group of self-selected bee-interested people. However, the study would be strengthened by further discussing on what is known about public knowledge of bee diversity, as it would help to justify the current study. Further, the author could make a stronger case for why public interest in and knowledge of native bees is important. Specifically, it is notable that the overwhelming majority of participants did not use scientific names. While the authors mention this and note that it is an issue, I’d like to see an expanded discussion of why this is significant, given the author’s previous work on this topic. For example, the public appears to have understood “blue-banded” bee to be a species whereas is represents part of a genus. This highlights the utility of using scientific vs. common names but also speaks to the lack of understanding of bee diversity more broadly. I thought it was interesting that there was such high visibility of the “blue-banded bees” and the article makes a good case for highlighting this group for conservation and scientific communication.

            Generally, the article is well-written, and is a nice case-study of the general lack of knowledge of bee diversity and taxonomy, even among interested persons. The study is not quantitatively rigorous (i.e. no statistical testing) but still provides valuable information about this phenomenon. The objectives are clearly stated and met, but the scope of the study is rather limited. However, the author is transparent about the limitations of the study. Thus, I refer to the editor’s discretion in determining whether the study meets the standard of the journal. Below I have highlighted specific areas I believe could be improved for future versions of the manuscript. In addition to these specific points, I would suggest strengthening the conclusions section a bit in the context of the study’s findings.

Line 59: where does this estimate originate (about there being hundreds of undescribed bees)?

Lines 67 – 71: consider splitting to multiple sentences, it is a bit of a run-on and hard to follow as written. Also some of these very short paragraphs can be combined.

Lines 89-90: please define what a gateway species is somewhere in the introduction before mentioning it here

Lines 86-88: Needs revising as it is grammatically incorrect, I think a word or two is missing. Could be written more concisely.  

Line 101: extra period should be deleted

Line 97-101: it’s not clear if the survey was given before the participants had access to the Facebook page or after they were able to see it.

Line 139: text says kleptoparasitic bees received 10 votes but figure only shows 9

Line 184: I would contend that the vast majority of people worldwide have very little knowledge of native bees. Are there other studies that have looked at public knowledge of bee diversity (or other insect taxa) that can be cited to expand on this point? If there is not much literature out there supporting this, then the case for your survey is even stronger as it puts data behind a phenomenon that many entomologists know well- the lack of public understanding of insects and their biodiversity.

Lines 207-210: This is a very good point!

237-243: Please expand on the economic importance of these bees. Being from another part of the world, I was not familiar with these species and had to look them up. There is an opportunity here to highlight some cool biology for people who are not familiar with these taxa.

265-270: Please revise it is too wordy as written.

Line 308 – change was to what

Line 323 – knowledgeable?

Line 323 – are to our?

Line 325 – there appears to be a missing word

Line 342: not to no?

Author Response

The study summarizes results from an online survey of people interested in wild bees. The author utilized a Facebook group called “The Buzz on Wild Bees” to ask joining members what their favorite Australian bee species is. The format allowed for open-ended answers. The survey results found limited taxonomic and biodiversity knowledge among survey takers. Of 400 people who provided answers only 19 provided a scientific name, and the majority provided a group rather than a species, underlying the lack of native bee knowledge among survey takers. The most popular group by far were the “blue banded” bees, indicating a potential flagship group for conservation. However, the three critically endangered bees in were not mentioned by any survey participants, highlighting a gap between public knowledge and conservation priorities. Further, the study identifies species or groups of species which were popular among participants, and explores possibilities for designating flagship species.

            The study is effective in communicating that native bee taxonomic knowledge is lacking among the Australian public, even among a group of self-selected bee-interested people. However, the study would be strengthened by further discussing on what is known about public knowledge of bee diversity, as it would help to justify the current study. Further, the author could make a stronger case for why public interest in and knowledge of native bees is important. Specifically, it is notable that the overwhelming majority of participants did not use scientific names. While the authors mention this and note that it is an issue, I’d like to see an expanded discussion of why this is significant, given the author’s previous work on this topic. For example, the public appears to have understood “blue-banded” bee to be a species whereas is represents part of a genus. This highlights the utility of using scientific vs. common names but also speaks to the lack of understanding of bee diversity more broadly. I thought it was interesting that there was such high visibility of the “blue-banded bees” and the article makes a good case for highlighting this group for conservation and scientific communication.

            Generally, the article is well-written, and is a nice case-study of the general lack of knowledge of bee diversity and taxonomy, even among interested persons. The study is not quantitatively rigorous (i.e. no statistical testing) but still provides valuable information about this phenomenon. The objectives are clearly stated and met, but the scope of the study is rather limited. However, the author is transparent about the limitations of the study. Thus, I refer to the editor’s discretion in determining whether the study meets the standard of the journal. Below I have highlighted specific areas I believe could be improved for future versions of the manuscript. In addition to these specific points, I would suggest strengthening the conclusions section a bit in the context of the study’s findings.

Response: Thank you for your review of the manuscript, I am glad you find it of value, and thank you for your recommendations, I have attended to each as detailed below.

Line 59: where does this estimate originate (about there being hundreds of undescribed bees)?

Response: I have included three references to support this statement now.

Lines 67 – 71: consider splitting to multiple sentences, it is a bit of a run-on and hard to follow as written. Also some of these very short paragraphs can be combined.

Lines 89-90: please define what a gateway species is somewhere in the introduction before mentioning it here

Lines 86-88: Needs revising as it is grammatically incorrect, I think a word or two is missing. Could be written more concisely.  

Line 101: extra period should be deleted

Line 97-101: it’s not clear if the survey was given before the participants had access to the Facebook page or after they were able to see it.

Line 139: text says kleptoparasitic bees received 10 votes but figure only shows 9

Line 184: I would contend that the vast majority of people worldwide have very little knowledge of native bees. Are there other studies that have looked at public knowledge of bee diversity (or other insect taxa) that can be cited to expand on this point? If there is not much literature out there supporting this, then the case for your survey is even stronger as it puts data behind a phenomenon that many entomologists know well- the lack of public understanding of insects and their biodiversity.

Response: I have now included a paragraph at the end of section 4.1. about findings on the lack of public understanding on bee biodiversity in other countries. This is the first study in Australia.

Lines 207-210: This is a very good point!

Response: Thank you!

237-243: Please expand on the economic importance of these bees. Being from another part of the world, I was not familiar with these species and had to look them up. There is an opportunity here to highlight some cool biology for people who are not familiar with these taxa.

Response: I have now included some information on the Meliponini, and extra references.

265-270: Please revise it is too wordy as written.

Response: I have split into two sentences, and revised (there was some repetition in the quote)

Line 308 – change was to what

Response: this has been changed.

Line 323 – knowledgeable?

Response: I am unsure what this is referring to.

Line 323 – are to our?

Response: I have rewritten this sentence.

Line 325 – there appears to be a missing word

Response: I am unsure what this is referring to.

Line 342: not to no?

Response: I have changed.

Reviewer 3 Report

Comments and Suggestions for Authors

The manuscript presentes an interesting and necessary approach rearding public knowledge about native bees. The study highlights the lack of knowledge among a public in Australia concerning bee taxonomy, which is essential information for effective biodiversity conservation actions.

Some importants points that, in my view, require adjustments:

  1. The table presented contains several issues, The first is that the sum of responses (last row) is duplicated, adding partial totals together with the individual responses.
  2. Still in Table 1, when scientific names are cited, I suggest placing them in italics (Amegilla cingulata, Amegilla (Asarapoda) bombiformis, ...)
  3. In the same table, the meaning of the abbreviations may not be obvious to non-Asutralian readers; I suggest indicating the full names of the abbreviations (AA, BB, TH, TC) in methodology or in table footnotes.
  4. I recommend clarifying the meaning of the Symbol “?” in the table.
  5. When comparing the data in Table 1 with the supplementary material, the association with the names not considered in the analysis is not straightforward. Therefore, I suggest that the table in the supplementary material include a new column indicating the names that were correctly identified as bees. It would also be useful to provide a clearer explanation or reference used to common names unequivocally attributed to the bee species (“blue tail” for example).

Figure 1 is quite interesting, as it shows the representativeness of each taxon along with a photogragh of the respective taxon. However, I beleive it could be improved. A graph type “donut chart” and larger photographs at the same size, would likely be more aesthetically appealing. It is not clear if the size of the image would be relative to the proportion in the responses; if it was the purpose, it is not clearly presented. In addition, I recommend adding scale bars to each photograph, providing a size reference for the bees. The count numbers for each response in the graph are unnecessary, as they are already included in the table. It would be better to connect the percentagem of responses directly to the corresponding image with guiding lines. 

Author Response

The manuscript presentes an interesting and necessary approach rearding public knowledge about native bees. The study highlights the lack of knowledge among a public in Australia concerning bee taxonomy, which is essential information for effective biodiversity conservation actions.

Some importants points that, in my view, require adjustments:

  1. The table presented contains several issues, The first is that the sum of responses (last row) is duplicated, adding partial totals together with the individual responses.

Response: I have revised.

  1. Still in Table 1, when scientific names are cited, I suggest placing them in italics (Amegilla cingulata, Amegilla (Asarapoda) bombiformis, ...)

Response: I have done so.

  1. In the same table, the meaning of the abbreviations may not be obvious to non-Asutralian readers; I suggest indicating the full names of the abbreviations (AA, BB, TH, TC) in methodology or in table footnotes.
  2. I recommend clarifying the meaning of the Symbol “?” in the table.

Response: I have removed the gender column (it was recommended not to discuss)

  1. When comparing the data in Table 1 with the supplementary material, the association with the names not considered in the analysis is not straightforward. Therefore, I suggest that the table in the supplementary material include a new column indicating the names that were correctly identified as bees. It would also be useful to provide a clearer explanation or reference used to common names unequivocally attributed to the bee species (“blue tail” for example).

Response: I have now done so regarding answers that were used in the analysis. I acknowledge that some had to be inferred as to which species/group they were referring to due to the lack of a clear system for common names. (blue tail isn’t a commonly known common name as far as I am aware, but it is unlikely to refer to something other than a “blue banded” Amegilla).

Figure 1 is quite interesting, as it shows the representativeness of each taxon along with a photogragh of the respective taxon. However, I beleive it could be improved. A graph type “donut chart” and larger photographs at the same size, would likely be more aesthetically appealing. It is not clear if the size of the image would be relative to the proportion in the responses; if it was the purpose, it is not clearly presented. In addition, I recommend adding scale bars to each photograph, providing a size reference for the bees. The count numbers for each response in the graph are unnecessary, as they are already included in the table. It would be better to connect the percentagem of responses directly to the corresponding image with guiding lines. 

Response:

I have now made the figure a donut chart, and have made the images approximately the same size, with % responses with lines coming out.

As these images are not taken with a microscope it is hard to add an accurate scale bar.

Round 2

Reviewer 1 Report

Comments and Suggestions for Authors

Thank you for adding paragraphs and providing further information! Bees are fascinating species and should be dealt with in biology classes. It's not just about Apis mellifera ...

Author Response

Thank you, I'm delighted you found this of interest, thank you for your feedback, I believe it strengthened the ms. Appreciate your time.